# Genome-Wide Identification and Expression Analysis of *YTH* Gene Family for Abiotic Stress Regulation in *Camellia chekiangoleosa*

**DOI:** 10.3390/ijms25073996

**Published:** 2024-04-03

**Authors:** Xiang Cheng, Sheng Yao, Jingjing Zhang, Dengbao Wang, Shaojun Xu, Qiong Yu, Kongshu Ji

**Affiliations:** 1State Key Laboratory of Tree Genetics and Breeding, Nanjing Forestry University, Nanjing 210037, China; chengxiang@njfu.edu.cn (X.C.); yaosheng0817@163.com (S.Y.); jjzhang@njfu.edu.cn (J.Z.); dbw@njfu.edu.cn (D.W.); jing_xsj11@163.com (S.X.); 2Key Open Laboratory of Forest Genetics and Gene Engineering of National Forestry & Grassland Administration, Nanjing 210037, China; 3Co-Innovation Center for Sustainable Forestry in Southern China, Nanjing Forestry University, Nanjing 210037, China

**Keywords:** epitranscriptomics, *Camellia chekiangoleosa* Hu, YTH domain, expression profiling

## Abstract

*N^6^*-methyladenosine (m^6^A) is essential for RNA metabolism in cells. The YTH domain, conserved in the kingdom of Eukaryotes, acts as an m^6^A reader that binds m^6^A-containing RNA. In plants, the YTH domain is involved in plant hormone signaling, stress response regulation, RNA stability, translation, and differentiation. However, little is known about the *YTH* genes in tea-oil tree, which can produce edible oil with high nutritional value. This study aims to identify and characterize the YTH domains within the tea-oil tree (*Camellia chekiangoleosa* Hu) genome to predict their potential role in development and stress regulation. In this study, 10 members of the *YTH* family containing the YTH domain named *CchYTH1-10* were identified from *C. chekiangoleosa*. Through analysis of their physical and chemical properties and prediction of subcellular localization, it is known that most family members are located in the nucleus and may have liquid–liquid phase separation. Analysis of cis-acting elements in the *CchYTH* promoter region revealed that these genes could be closely related to abiotic stress and hormones. The results of expression profiling show that the *CchYTH* genes were differentially expressed in different tissues, and their expression levels change under drought stress. Overall, these findings could provide a foundation for future research regarding *CchYTHs* in *C. chekiangoleosa* and enrich the world in terms of epigenetic mark m^6^A in forest trees.

## 1. Introduction

In the regulation of RNA production and degradation, scientists have introduced the concept of epigenetics. Epigenetics is reversible, heritable change in gene expression that occurs without altering the DNA sequence. Currently, most people believe that epigenetic modification, like DNA, is an important factor in determining gene expression and individual phenotypes. Numerous studies have also shown that they are closely related to biological development and disease [1,2]. In 2011, Professor Chuan He and his team from the University of Chicago discovered FTO, the first m^6^A demethylase, revealing the reversible modification of m^6^A [3]. After that, a relatively clear m^6^A-modified map was gradually constructed with the efforts of many scientists, which contains three core regulatory factors: writer, eraser, and reader.

The *YTH* (YT521-B homology) domain includes RNA-binding domains and belongs to the *PUA* superfamily. It is speculated that the YTH domain is particularly abundant in plants [4]. The predicted secondary structure of the YTH domain consists of four α-helices and six β-strands, and, via its conserved amino acid sequences, shows the ability to bind with single-stranded mRNA [5]. The gene that carries this domain was first discovered in 1999, and it was cloned from the astrocytes of rats (*Rattus norvegicus*), with its expression upregulated in reoxygenated astrocytes [6]. The *YTH* gene family has been reported as a representative of reader, which has been divided into five subfamilies: YTHDC1, YTHDC2, and YTHDF1–3, based on the main sequences and domain tissueization of five YTH domain proteins found in humans [7]. More and more studies have found that proteins containing the YTH domain play a role in multiple aspects of plant life, such as developmental progress and responses to abiotic and biotic stress. Moreover, the *YTH* gene family has been gradually identified in many plants, such as wheat [8] and *Citrus sinensis* [9].

Previous studies on YTH proteins have predominantly focused on humans and animals. Among them, YTHDF1 and YTHDF3 have been shown to promote protein translation by interacting with initiation factors and ribosomes [10]. YTHDF2, YTHDF3, and YTHDC2 share a similar function in degrading mRNA [11]. YTHDC1 is associated with mRNA splicing and nucleation [12]. In many plants, genes containing the YTH domain can also respond to various abiotic stresses, including high salinity, drought, heat, cold, and polyethylene glycol stresses [13,14]. For example, in dicot species *Arabidopsis thaliana*, the YTHDFA subfamily members *ECT2/3/4* are important for cell proliferation, plant organogenesis, and nitrate transport [15]. Overexpression of *MhYTP1* or *MhYTP2* makes plants more sensitive to NaCl but more resistant to nutrient deficits [16]. Knocking down *OsYTH*s led to downregulation of many biological regulatory genes and resulted in growth defects in rice. In addition, functionally deficient *OsYTH* mutants become more tolerant to salt stress but more sensitive to other abiotic stresses (including response to drought, cold, heat, ABA, ion deficiency, and UV damage) [17]. These findings demonstrate that YTH proteins acting as m^6^A readers can medicate the function of m^6^A in response to environmental stress throughout the lifetime of plants.

*Camellia chekiangoleosa*, collectively known as “oriental olive oil”, is naturally distributed in southern China. This kind of camellia tree thrives in warm and damp conditions but cannot tolerate cold or drought, preferring weak light. *C. chekiangoleosa* is also an industrial raw material for soaps, lubricants, and pharmaceuticals [18]. However, its artificial cultivation history is relatively short [19], and early research on *C. chekiangoleosa* mainly focused on germplasm resource conservation, genetic diversity analysis, molecular marker-assisted breeding, and other aspects [20,21,22]. Although increasing studies on m^6^A readers have been reported regarding various plants in recent years, little is known about m^6^A readers such as the YTH domain-containing RNA-binding protein family in *C. chekiangoleosa*. In our study, we employed the conserved domain to identify the *YTH* gene family in *C. chekiangoleosa* and analyzed the basic physical and chemical properties, subcellular localization, and tissue-expression pattern. Given that *C. chekiangoleosa* prefers moisture and that many cis-elements related to drought stress have been found by cis-acting element analysis, we conducted an expression pattern analysis under drought stress. These results not only further validate previous studies but also enrich our understanding of the *YTH* gene family in plants. Furthermore, our study offers insights regarding the artificial cultivation and genetic improvement of *C. chekiangoleosa*.

## 2. Results

### 2.1. Genome-Wide Identification, Basic Physicochemical Properties, and Phylogenetic Analysis of CchYTH Genes

Searching against the *C. chekiangoleosa* genome resulted in identification of 10 *YTH* genes. The *YTH* family members are named *CchYTH1–CchYTH10* according to the sequence in which they appear on the chromosomes. The number of amino acids in the YTH family ranges from 386 to 1356, with an average of 681.8. Their predicted molecular weights range from 150.028 kDa to 50.951 kDa, with an average of 75.603 kDa. The isoelectric points (pI) range from 5.35 to 9.22, with an average of 6.21. Among the 10 YTH proteins, 40% are predicted to be stable proteins, and 60% are predicted to be unstable. The theoretical instability index ranges from 29.17 to 55.16. Six of them are unstable proteins (CchYTH3, CchYTH4, CchYTH6, CchYTH7, CchYTH8, and CchYTH10), with instability index values higher than 40. The prediction results of gene subcellular localization indicate that seven (70%) of the ten YTH gene family members are localized to the nucleus and three (30%) localized in the cytoplasm (Table 1).

In our study, the YTH proteins of *C. chekiangoleosa* are divided into DC and DF families based on the research of the YTH proteins in other plants. The DF group has eight members (CchYTH1–6, CchYTH8, and CchYTH9). The DC group has two members (CchYTH7 and CchYTH10). Then, one-hundred-fifteen proteins from eleven species are combined with YTH proteins from *C. chekiangoleosa* to construct an evolutionary tree and further divided into five subfamilies. There is one member in the DFA subfamily (CchYTH9), two members in the DFB subfamily (CchYTH3 and CchYTH4), and the largest number of members in the DFC subfamily (CchYTH1, CchYTH2, CchYTH5, CchYTH6, and CchYTH8). There is one member in the DCA subfamily (CchYTH7) and one member in the DCB subfamily (CchYTH10) (Figure 1).

### 2.2. Chromosomal Localization Prediction of CchYTH Genes

*C. chekiangoleosa* has 15 pairs of chromosomes. The genes encoding the 10 *CchYTH* gene family members are an uneven distribution of chromosomes 1, 2, 3, 5, 7, 8, 9, 12, and 13 of *C. chekiangoleosa*. Except for chr02, there is only one gene on each chromosome. There is no positive correlation between the length of chromosome and the number of *CchYTH* genes. No genes in the *CchYTH* gene family are clustered into one tandem duplication event region on any chromosome. The unbalanced distribution of *CchYTH* on chromosomes indicates that *C. chekiangoleosa* has undergone genetic variation during the process of evolution (Figure 2).

### 2.3. Collinearity Analysis of the YTH Family in Camellia chekiangoleosa

After, a collinearity analysis of the *YTH* genes between *C. chekiangoleosa*, *A. thaliana*, and *P. trichocarpa* was conducted. We further elucidated the phylogenetic mechanism of *CchYTHs* (Figure 3). The results that can be drawn from observation of collinearity indicate that a total of nine *YTH* genes in *P. trichocarpa* exhibit a collinear relationship with six *YTHs* in *C. chekiangoleosa*. However, in *A. thaliana*, only six *ECT* genes are collinear with five *CchYTH* genes (Figure 3). Consequently, the *CchYTH*s demonstrate a higher frequency of collinear gene pairing with woody plants compared to herbaceous plants. Our results reveal that the *YTH* gene family is conservative and withstands relatively strong selection pressure during plant evolution.

### 2.4. Gene Structure, Conserved Domain, and Motif Analysis of CchYTH Genes

We have analyzed the exon–intron structure of the *CchYTH* genes based on their evolutionary classification. All the *CchYTH* genes contain introns, and the majority of these genes contain between five and nine introns. The number of exons and UTRs significantly varies across all the *CchYTH* genes, indicating putative distinguished functions among these CchYTH proteins (Figure 4a).

According to the motif analysis, a total of 10 motifs were found in these CchYTH proteins. All the CchYTH proteins contained motif 1, motif 6, and motif 10. Motif 2 is only present in the DF subfamily (Figure 4b), while motif 7 exists in the DC subfamily. In all the CchYTH proteins, the YTH domain is the only recognizable module at their C-terminus that is similar to those of other species. Notably, only *CchYTH9* of the DCA family has a special zinc finger (CCCH) domain (Figure 4c).

Additional multiple sequence alignment of CchYTH proteins shows that many functional sites were conserved and displayed a conserved aromatic cage. Notably, the aromatic cage of the YTH domain of all the DF subfamilies consists of tryptophan residues (WWW), while the second tryptophan of the DCB subfamily member is replaced by serine (S) (Figure 4d). Moreover, the N-termini of the CchDF and CchDCB proteins possess low-complexity regions containing Y/P/Q-rich regions; however, the Y/P/Q-rich region in CchDCA was located between the zinc finger repeat (CCCH domain) and the YTH domain (Figure 4e; Appendix A).

### 2.5. Most of the CchYTH Proteins May Participate in the Process of Liquid–Liquid Phase Separation

Prion-like Amino Acid Composition (PLAAC) has been shown to be a powerful tool for predicting prion-like domains (PrLDs) in proteins. It has been used to identify PrLDs in a wide range of proteins, including those involved in RNA binding, transcriptional regulation, and signal transduction [23]. According to the predicted results, most of the CchYTH proteins (CchYTH1-9) are predicted to contain one or two highly disordered PrLDs. The PrLD domain has been shown to mediate liquid–liquid phase separation of proteins. This suggests that CchYTH proteins may undergo phase separation in a manner similar to the human YTHDF1–YTHDF3 proteins, further supporting their liquid-like properties (Figure 5; Appendix A).

### 2.6. Tissue-Specific Expression Profiling of CchYTHs

To determine the possible physiological role of *CchYTH* in growth and development, we analyzed the tissue-specific expression patterns of the *CchYTH* genes in five different tissues (root, stem, leaf, flower, and terminal bud) by RT-qPCR. The results show that the expression patterns of *CchYTH* genes vary across the five tissues. As shown in Figure 6, the *CchYTH* genes are expressed constitutively in all the tissues tested, but their expression patterns are different. *CchYHT3* and *CchYTH5* are highly expressed in leaves but at low levels in roots, stems, flowers, and terminal buds. The expression level of *CchYTH2* is high in flowers but low in other tissues, the same for the expression levels of *CchYTH7*, *CchYTH8*, and *CchYTH9*. The expression level of *CchYTH6* is elevated in stems. In general, the expression levels of *CchYTH* genes are the lowest in roots, while the expression levels of five *CchYTH* genes (*CchYTH3*, *4*, *5*, *8*, and *9*) among these ten *CchYTH* genes are higher in leaves (Figure 6). Furthermore, we analyzed the expression levels of these 10 genes across all five tissues. It is worth noting that the expression of *CchYTH9* is significantly higher than that of other genes (Figure 6).

### 2.7. Subcellular Localization of CchYTH9

The subcellular localization prediction results show that most YTH family proteins are located in the nucleus. In order to verify their accuracy, we selected *CchYTH9*, which is predicted to be located in the nucleus and has the highest expression in all the tissues we analyzed, for subcellular localization experiments. The results (Figure 7) show that the eGFP fluorescent signal originates from the nucleus, and there is no fluorescent signal in the cytoplasm or cell membrane, indicating that the protein works in the nucleus, consistent with the previously predicted subcellular localization.

### 2.8. Cis-Acting Elements Analysis of the 10 CchYTH Genes Promoter

A cis-acting element is present in the sequence on the side of the gene that can affect gene expression. The cis-acting elements include promoters, enhancers, regulatory sequences, and inducible elements that participate in the regulation of gene expression. By predicting the cis-acting elements of the sequence of 2000 bp promoter regions upstream of *CchYTH*s, we found that the *CchYTH* promoter region contains multiple stress-responsive and hormonal response-related elements, such as light responsiveness element (Sp1; G-box), low-temperature responsiveness element (LTR), drought element (MBS), auxin responsiveness element (AuxRR; TGA-element), salicylic acid responsiveness element (TCA-element), gibberellin-responsive element (GARE-motif; P-box; TATC-box), and so on (Figure 8a). After counting the numbers of these cis-acting elements in each gene, we found that drought elements were the major cis-acting elements among these components, which can be found in most genes. Furthermore, drought stress has the greatest impact on the artificial cultivation of safflower *C. chekiangoleosa*. Therefore, it is worth studying the impact of these abiotic stresses on *CchYTH* family genes (Figure 8b).

### 2.9. Changes in Relative Expression of the CchYTH Genes under Drought Stress

After 12 days of continuous drought treatment, most of the *CchYTH* genes are involved in the regulation of drought. Among them, four members (*CchYTH2*, *CchYTH3*, *CchYTH5*, and *CchYTH10*) reach the highest relative expression level after 3 days of treatment, and another four members (*CchYTH1*, *CchYTH6*, *CchYTH7*, and *CchYTH9*) reach the highest relative expression level after 6 days of treatment. Only two genes show no change (*CchYTH4*) or even a decrease (*CchYTH8*) under drought treatment. In general, with the increase in treatment days, the relative expression of most *CchYTH* genes shows a trend of first increasing and then decreasing (Figure 9).

## 3. Discussion

*N^6^*-Methyladenosine (m^6^A) is one of the most common post-transcriptional modifications in eukaryotic RNAs, which plays a crucial role in gene regulation and the maintenance of genome stability [18]. The recognition of m^6^A by its readers is essential for RNA metabolism, such as decay, stability, folding, and translation. Many m^6^A reader proteins have been identified, and most possess a YT521-B homologous (YTH) domain with an aromatic cage that specifically recognizes the GG (m^6^A) C sequence [24]. In this study, 10 *CchYTH* genes distributed on nine chromosomes were identified from the genome of *C. chekiangoleosa*. Compared with previous studies, there were 13 *YTH* genes in *Arabidopsis thaliana* [25], 12 *YTH* genes in rice [26], 26 *YTH* genes in apple [27], and 10 *YTH* genes in citrus. These different distributions of *YTH* genes between species suggest that the m^6^A-YTH regulatory mechanism might vary widely in various plants. Therefore, it is worthwhile to study the distinctive function of YTH proteins in *C. chekiangoleosa*.

Through the analysis of physicochemical properties and gene structure of 10 CchYTH proteins, it was found that the protein length, predicted molecular weights, pI, and the number of exons and UTRs of CchYTHs varied significantly in *C. chekiangoleosa*, indicating putative distinguished functions among these CchYTH proteins. Furthermore, the majority of *CchYTH* gene family members were predicted to be localized to the nucleus, which was then verified by the localization of CchYTH9 in the nucleus of tobacco, suggesting unique functions of each CchYTH protein and the supposed distinctive function network of m^6^A readers in *C. chekiangoleosa* compared to other plants [9]. The Prion-like Amino Acid Composition prediction results indicate that almost all the members of the *YTH* family except CchYTH10 possess the PrLDs domain, the prerequisite for the protein to have the ability of liquid–liquid phase separation, indicating that CchYTH proteins may participate in the liquid–liquid phase separation process like in mammals or *A. thaliana* to determine the fate of m^6^A-modified RNA, thereby regulating vital life processes [28,29,30,31,32]. No such research has been reported in forest trees yet, so it is worthwhile to study CchYTH proteins from a phase separation perspective.

The primary structure of subpopulations is closely related to the biological function of proteins. Therefore, the identification of plant homologous genes based on phylogenetic relationships could help to predict gene function [33,34]. To further study the evolutionary relationships among CchYTH proteins and obtain a detailed classification of these genes in *C. chekiangoleosa*, phylogenetic trees were constructed within 10 representative species. Through the study of this paper, we know that *CchYTH* genes can be divided into two major subfamilies, including eight DF subfamily members and two DC subfamily members; and DF could be further divided into three subfamilies (DFA, DFB, and DFC), while DC could be further divided into two subfamilies (DCA and DCB), similar to previous research in other plants [35]. However, the numbers of the members in these subfamilies across various plants are quite different. For example, there were two *YTHDC* and eleven *YTHDF* genes in *Arabidopsis thaliana* [25], one *YTHDC* and eleven *YTHDF* genes in rice [26], and four *YTHDC* and twenty-two *YTHDF* genes in apple [27]. These different distributions of the subfamily genes between species further suggest that, aside from possessing the same functions as other plants, members of the *CchYTH* family may have special functions in the m^6^A-YTH regulatory mechanism that are worth studying in *C. chekiangoleosa*.

According to the results from collinearity analysis of *YTH* genes between *C. chekiangoleosa*, *A. thaliana*, and *P. trichocarpa*, it has been verified that the *YTH* gene family is conservative and withstands relatively strong selection pressure during plant evolution. The collinearity of *CchYTH* with poplar is greater than that with *Arabidopsis thaliana*, implying that the evolutionary similarity between YTH proteins in woody plants is higher with smaller divergence. This discovery suggests the necessity to study the *YTH* genes in woody plants beyond herbaceous plants.

It has been reported that the conserved domains of genes in the *YTH* gene family are basically the same [36]. Through the analysis of the conserved domain and motif, we found a conserved YTH domain in the C-terminus of *CchYTH* genes consistent with those of the *YTH* genes in *Arabidopsis thaliana*, *Oryza sativa*, and other species [25,26]. Notably, only *CchYTH9*, which is classified into the DCA family, has a special zinc finger (CCCH) domain (Figure 4c). Most of the aromatic cage of the YTH domain consists of tryptophan residues (WWW) [17]. Additional multiple sequence alignment of CchYTH proteins shows that the aromatic cage of the YTH domain of all the DF subfamilies consists of tryptophan residues (WWW), while the second tryptophan of DCB subfamily member chYTH10 is replaced by and serine (Figure 4d). This comparison of amino acid sequences in conserved domains shown that the result was different from previous reports, suggesting that the DC subfamily members of *C. chekiangoleosa* have undergone some changes in structure, resulting in a special function during evolution. In summary, the YTH domain is sufficient and necessary to identify the YTH gene, and the CCCH domain can distinguish DCA subfamily members (*CPSF30-L)* from other family members. However, unlike the DF subfamily, the second tryptophan of its DCB subfamily members is replaced by serine (S), which might be the key site leading to the functional difference regarding these family members in *C. chekiangoleosa*.

According to previous studies on the function of YTH family genes in *A. thaliana*, *P. pilocarpa*, and other plants, the *YTHDF* and *YTHDC* subfamily genes play an indispensable role in the translation of m^6^A-modified mRNA, thus affecting the growth and development of plants [37,38,39]. A comprehensive gene expression analysis of YTH gene family members revealed that the *YTH* genes exhibited similar tissue-expression patterns across several plant species [40]. For example, in *C. sinensis* and common wheat, most *YTH* genes are highly expressed in leaves, flowers, and fruits but low in roots [8,9]. Our study also demonstrated similar tissue-expression levels of most *CchYTH* genes in *C. chekiangoleosa*, suggesting that, like most plants, the CchYTH proteins play an important role in the flourishing sites of nutritional and reproductive growth. Drought is the major environmental stress that affects plant growth and development and ultimately crop yields [41,42]. Regarding the major environmental threats for *C. chekiangoleosa* and the dominant enrichment of drought-related cis-acting elements in the promoter of *CchYTH* genes, we investigated the expression of *CchYTH* genes under drought stress to study the possible regulatory mechanism of m^6^A through the YTH reader in response to environmental stress. After being drought-treated, we found that most of the *CchYTH* genes were upregulated by drought stress, indicating that the *CchYTH* gene family may be beneficial for the growth of *C. chekiangoleos* during drought stress. In general, our results provide the foundation for the functional analysis of *CchYTHs* and genetic improvement of *C. chekiangoleosa* in the future.

## 4. Materials and Methods

### 4.1. Genome-Wide Identification of Camellia chekiangoleosa YTH Genes

Genome-wide data for *C. chekiangoleosa* were found and downloaded on the website National Genomics Date Center (https://ngdc.cncb.ac.cn/ accessed on 15 December 2023). To accurately locate all members of *YTH* gene family in *C. chekiangoleosa*, the amino acid sequences of *YTH* genes in *Arabidopsis thaliana* were downloaded from TAIR (http://www.arabidopsis.org/index.jsp accessed on 16 December 2023) and blasted the candidate target proteins by using the sequences of *AtYTH* gene family members as templates. Downloaded the hidden Markov model (HMM) of the YTH domain (PF04146) from the Sanger database (http://pfam.xfam.org/family/PF04146 accessed on 18 December 2023), and then HMMER 3.0 software was used to query the genome database of *C. chekiangoleosa*; the candidate sequence is obtained again at the end. Redundant sequences were removed, and the remaining putative YTH protein sequences were subjected to analyses by their YTH domain. Finally, we used InterPro (http://www.ebi.ac.uk/interpro accessed on 15 December 2023) to verify the integrity of YTH domain.

### 4.2. Basic Physicochemical Properties, Amino Acid Sequence Alignment, and Phylogenetic Analysis of Camellia chekiangoleosa YTH Genes

We used Expasy-ProtParam tool (https://web.expasy.org/protparam/ accessed on 16 December 2023) to analyze their basic physical and chemical properties. Multiple protein sequence alignments were performed using ESPript 3.0 (https://espript.ibcp.fr/ accessed on 22 December 2023) and DNAMAN. After that, we used Wolf PSORT (https://wolfpsort.hgc.jp/ accessed on 20 December 2023) and several other online tools to predict the subcellular localization of these 10 genes.

In order to understand the evolutionary history of the *YTH* gene family in *C. chekiangoleosa*, we compared 105 YTH-containing proteins from 10 representative YTH domains including *Arabidopsis thaliana*, *Eucalyptus grandis*, *Glycine max*, *Vitis vinefera*, *Citrus sinensis*, and others with YTH domain in *C. chekiangoleosa* (Appendix A). The phylogenetic tree was then constructed using MEGA 11 and iTOL (https://itol.embl.de/ accessed on 17 December 2023).

### 4.3. Chromosomal Localization Prediction and Naming of Members of YTH Family in Camellia chekiangoleosa

The *C. chekiangoleosa* genome annotation GFF file was downloaded from the NSTI website (https://ngdc.cncb.ac.cn/ accessed on 15 December 2023). The gene locations visualized from the GTF/GFF module were input into the GFF file along with the gene ID of *YTH* family members. These were used to draw the chromosome location map by TBtools. Then, each gene was named based on its location on the chromosome.

### 4.4. Duplication Events of CchYTH Genes

To gain insights into the evolution of *CchYTH* genes, we investigated genome duplication events in this gene family. We downloaded the genomes and annotated files of *A. thaliana* and *P. trichocarpa* from NCBI (National Center for Biotechnology Information nih.gov). Then, we used MCScanX (http://chibba.pgml.uga.edu/mcscan2/#tm accessed on 16 December 2023) to perform collinear analysis and plot.

### 4.5. Structure, Conserved Domain, Motif, and Promoter Area Cis-acting Element Analysis of CchYTH

Using the *C. chekiangoleosa* genome annotation file from the National Genome Data Center (https://ngdc.cncb.ac.cn/ accessed on 14 December 2023), we analyzed and predicted exons and introns of ten members of the *CchYTH* family. Then, we used TBtools to visualize the structure of *CchYTH* family members. The conserved motifs of 10 YTH protein sequences of *C. chekiangoleosa* were analyzed using the online tool MEME (https://meme-suite.org/meme accessed on 25 December 2023). The conserved domains of 10 *C. chekiangoleosa* YTH protein sequences were analyzed using the Web CD-Search tool in NCBI (https://www.ncbi.nlm.nih.gov/ accessed on 18 December 2023). Then, we visualized the analysis results of conserved domains and conserved motifs by TBtools.

To investigate the cis-acting elements and their functions regarding individual members of the CchYTH gene family, we used PlantCARE (https://bioinformatics.psb.ugent.be/webtools/plantcare/html/ accessed on 19 December 2023) to predict the cis-acting elements of the sequence of 2000 bp promoter regions upstream of CchYTH online. After that, we used Origin and TBtools to draw a heat map to count the number of elements associated with abiotic stress.

### 4.6. Prediction of Liquid–Liquid Phase Separation

We used PhaSePred (http://plaac.wi.mit.edu/ accessed on 18 December 2023), a phase separation protein prediction tool developed by Tingting Li [43], to predict the liquid–liquid phase separation of CchYTHs. The predicted results are visualized using AI.

### 4.7. Plant Materials and Treatments

Two-year-old *C. chekiangoleosa* seedlings, grown by the Key Laboratory of Forest Tree Genetic Breeding at Nanjing Forestry University, were used for studying the expression level of CchYTHs in different tissues (roots, flowers, leaves, stems, and terminal buds). We collected three biological replicates from each tissue, which were then immediately frozen in liquid nitrogen and stored at −80 °C. Drought stress treatment was performed according to the previous report [44]: 2-year-old plants in pots (30 × 20 × 22 cm) were withheld watering for 12 days, while the plants under normal conditions were watered normally. Leaf samples were collected from each treated seedling at 0 d, 4 d, 8 d, and 12 d after the stress was applied, along with untreated leaf samples at 0 h as a control group. The collected samples were immediately frozen in liquid nitrogen and stored at −80 °C for subsequent RNA extraction.

### 4.8. RNA Extraction and RT-qPCR

In order to analyze the expression pattern of *CchYTH*, real-time quantitative PCR (RT-qPCR) was used to study the expression of *CchYTH* family members in different tissues and under drought treatment of two-year-old *C. chekiangoleosa* cultured in nutritious soil under conditions of 25 °C and 16 h/8 h light and dark cycle in the Key Laboratory of Forest Tree Genetic Breeding at Nanjing Forestry University. We extracted RNA from the annual roots, stems, leaves, flowers, and terminal buds of *C. chekiangoleosa* by Trizol method [45]. The extracted RNA was then analyzed using 1% agarose gel electrophoresis to confirm its quality and integrity. The RNA was reverse transcribed to synthesize the first strand using a Hifair^®^ Ⅲ 1st Strand cDNA Synthesis Super Mix (Yeasen, Shanghai, China). The cDNA was diluted tenfold. Primers were designed based on the sequences of *C. chekiangoleosa* in the CDS database using Primer 5 software (Appendix A). *CchActin* was used as a reference control gene. Hieff UNICON^®^ Universal Blue qPCR SYBR Green Master Mix (Yeasen, Shanghai, China) was used as the reaction enzyme mixture. Each PCR mixture (20 μL) contained 2 μL of cDNA (10-fold dilution), 10 μL of SYBR Green Real time PCR Master Mix, 0.4 μL of each primer, and 7.2 μL of ddH_2_O. The qRT-PCR reaction was carried out under the following conditions: 1 cycle at 98℃ for 3 min, followed by 40 cycles at 95 °C for 15 s, 60 °C for 30 s, and 72 °C for 30 s. Using the RT-qPCR results of *CchYTH1* in roots as control, a relative expression calorimetric was created with TBtools. As for the analysis of relative expression of drought stress, we used the RT-qPCR results of 0 d without any treatment as control. The 2^−∆∆Ct^ method was used to evaluate genes expression levels [46]. Relative expression histograms were created with GraphPad Prism 8.

### 4.9. Subcellular Localization of CchYTH9 Protein

The coding sequences of the *CchYTH9* gene amplified by using two primers: CchYTH9-F and CchYTH9-R (Appendix A) was inserted into pCAMBIA1305 vector between XbaI and SalI restriction sites with C-terminal eGFP, driven by the CaMV35S promoter. The vectors carrying p35S:: CchYTH9-eGFP and p35S:: eGFP were introduced into Agrobacterium cells EHA105. The tobacco leaves (grown for 20 d) were infested by *Agrobacterium*-mediated instantaneous transformation. Leaf samples (0.5 × 0.5 cm) dyed with DAPI were observed under a laser confocal microscope (Zeiss LSM710 META, UV excitation wavelength 488 nm, Carl Zeiss AG, Germany) at an excitation wavelength of 488 nm for GFP and 405 nm for DAPI [47].

### 4.10. Statistical Analysis

For statistical analysis, IBM SPSS Statistics 25 software was used. One-way ANOVA was used to compare the differences between means. The gene expression during 0 d stress treatment and CchYTH1 expression in roots was used as a control for significant analysis. Statistically significant difference was considered at * *p* < 0.05 ** *p* < 0.01 *** *p* < 0.001 **** *p* < 0.0001.

## 5. Conclusions

In our study, 10 *CchYTH* genes containing complete YTH conserved domain were identified from *C. chekiangoleosa*. According to the phylogenetic tree, the conserved domain, and motif analysis, these genes were divided into five subfamilies (DFA, DFB, DFC, DCA, and DCB). Through analysis of the organization of the proteins and prediction of subcellular localization, the distinct motif distribution and domain architectures were observed in the *CchYTH* gene family, and most family members were located in the nucleus, with the ability to trigger liquid–liquid phase separation. All of the aromatic cage pocket of CchYTHDF is composed of tryptophan, tryptophan, and tryptophan (WWW). In contrast, this pocket is composed of tryptophan, tryptophan, and l-Threonine (WWS) in CchYTHDCB protein. The results of expression profiling showed that the *CchYTH* genes were differentially expressed in different tissues. The existence of abiotic stress-related cis-acting elements in the *CchYTH* promoter region indicated that these genes could be involved in abiotic stress response, especially drought stress. Further drought stress experiments confirmed that their expression levels indeed changed under drought stress. In conclusion, this identification provides a theoretical framework and prospective direction for the application of RNA epitranscriptomic engineering on *C. chekiangoleosa* breeding in the future.

## Figures and Tables

**Figure 1 ijms-25-03996-f001:**
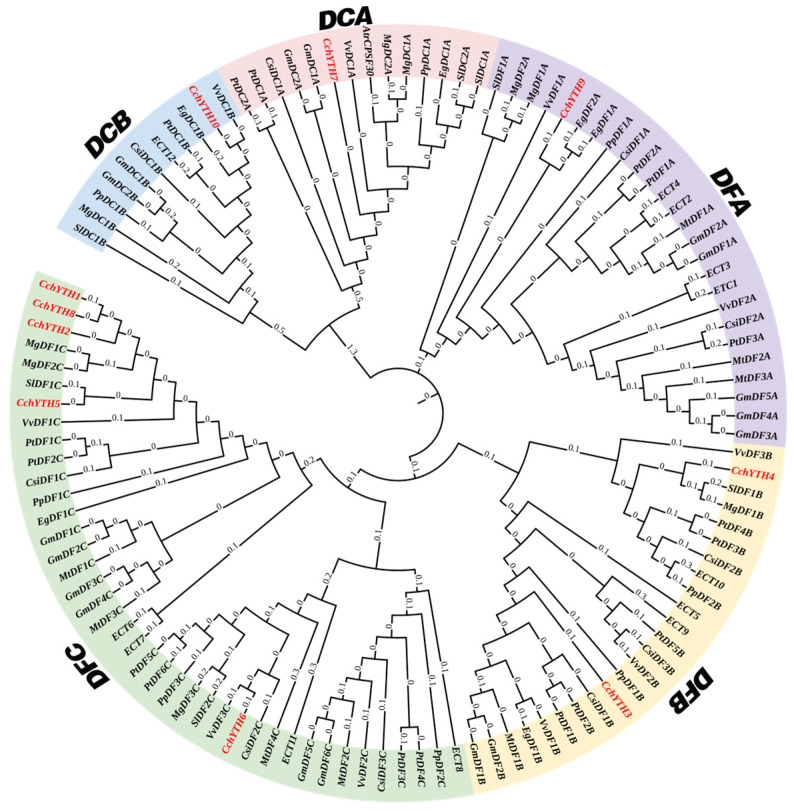
Phylogenetic analysis of YTH proteins in *C. chekiangoleosa*. Phylogenetic relationship between YTH domain-containing proteins (YTHs) identified from 10 plant species. YTH family members were clustered into two groups and further divided into two and three subgroups, respectively (see Appendix A for YTH sequences). The phylogenetic tree was constructed using MEGA 11 by the neighbor-joining method and iTOL. Bold text represents each subgroup, and the *C. chekiangoleosa* YTH protein is represented by a red letter.

**Figure 2 ijms-25-03996-f002:**
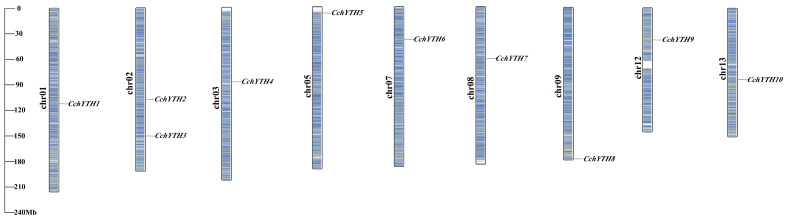
Chromosomal locations of the identified *CchYTH*s in *C. chekiangoleosa*. The size of a chromosome is expressed by its relative length. The chromosome numbers are shown at the left side of each chromosome and marked in bold boldface. Scale bar on the left indicates the chromosome lengths (Mb).

**Figure 3 ijms-25-03996-f003:**
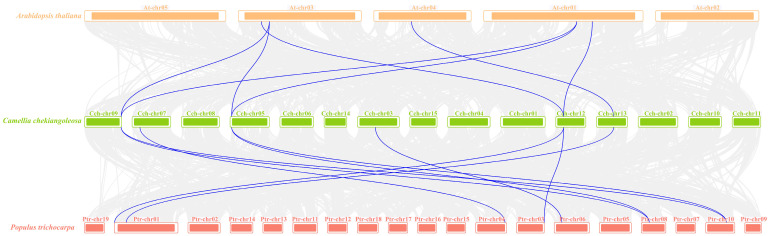
Synteny analysis of YTH genes between *A. thaliana* (orange) and *C. chekiangoleosa* (green) and *P. trichocarpa* (red). Gray lines in the background indicate the collinear blocks within the genomes of *C. chekiangoleosa* and other plants, and the blue lines indicate the syntenic *YTH* gene pairs.

**Figure 4 ijms-25-03996-f004:**
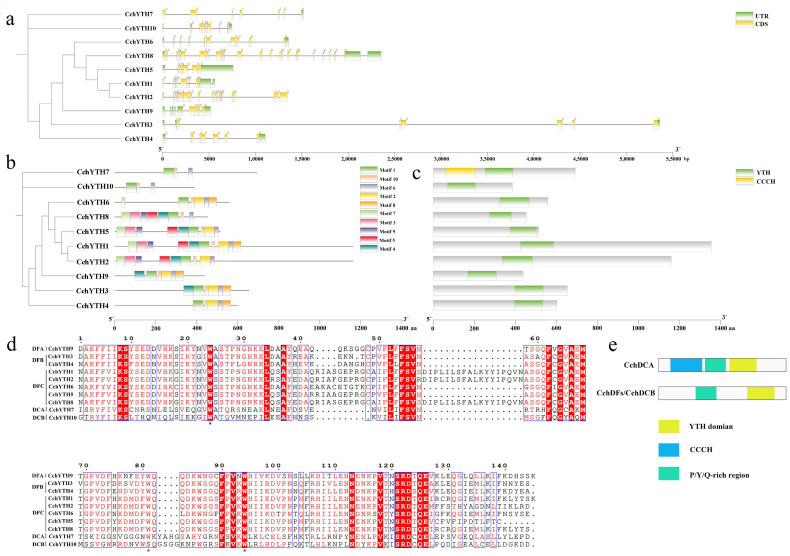
Domains, exon–intron, and motif structures of CchYTHs. (**a**) Yellow box, green box, and black line indicate exon, UTR, and intron, respectively. The phases of intron (0, 1, and 2) are shown on top of the corresponding introns. (**b**) Conserved motifs of *CchYTH*s. Different motifs are distinguished by different colors. (**c**) Conserved domains of *CchYTH*s. (**d**) Sequence alignments of YTH domain in CchYTH family proteins. Asterisks indicate the tryptophan position. (**e**) Locations of Y/P/Q-rich regions in CchDFs and CchDCs.

**Figure 5 ijms-25-03996-f005:**
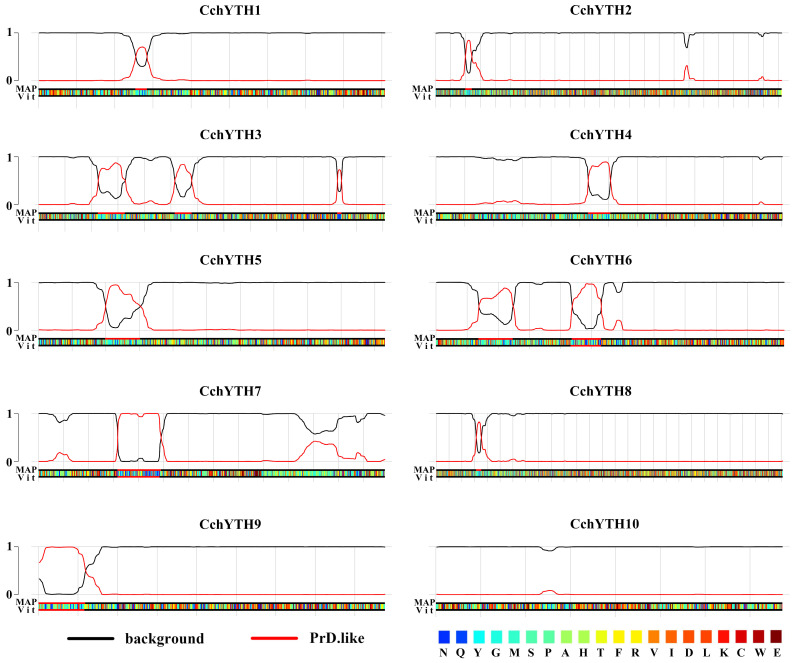
Predictions of PrLDs and disordered regions created by the Prion-like Amino Acid Composition (PLAAC; http://plaac.wi.mit.edu/ accessed on 12 December 2023) using a hidden Markov model (HMM) algorithm. The black line represents the background, and the red line is the prediction of the prion structure region. If the red line is in the non-baseline region, it indicates that the prion structure region is at that location and the phase transition is highly likely.

**Figure 6 ijms-25-03996-f006:**
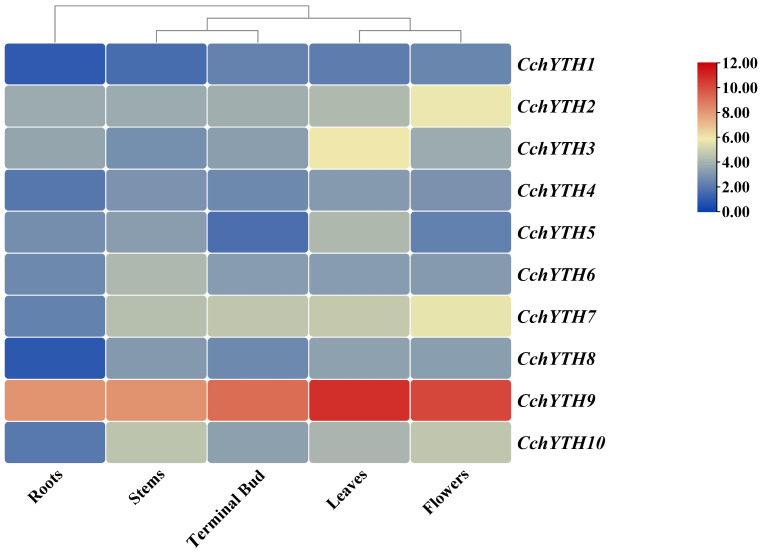
Heat map of tissue expression specificity analysis of 10 CchYTHs. The relative expression is calculated with reference to the expression of CchYTH1 in the root, and the data are normalized by Log Scale in the heat map. The horizontal coordinate indicates different tissues, and the ordinate represents gene names. Red box indicates high expression; yellow box indicates medium expression; blue box indicates low expression.

**Figure 7 ijms-25-03996-f007:**
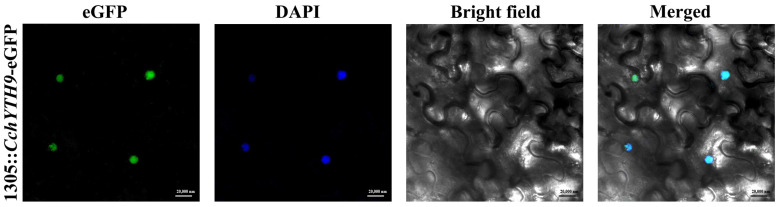
Subcellular localization analysis of *CchYTH9*. The 1305::*CchYTH9*-eGFP was transiently expressed through agroinfiltration in *Nicotiana benthamiana* leaves, and green fluorescence of the eGFP was viewed using confocal laser microscopy (first on the left). DAPI is an embedder containing specific AT sequence DNA, which can make the nucleus fluoresce blue for a short period of time (second on the left). The same cells were also viewed by transmission microscopy (second on the right), and three images were merged (first on the right).

**Figure 8 ijms-25-03996-f008:**
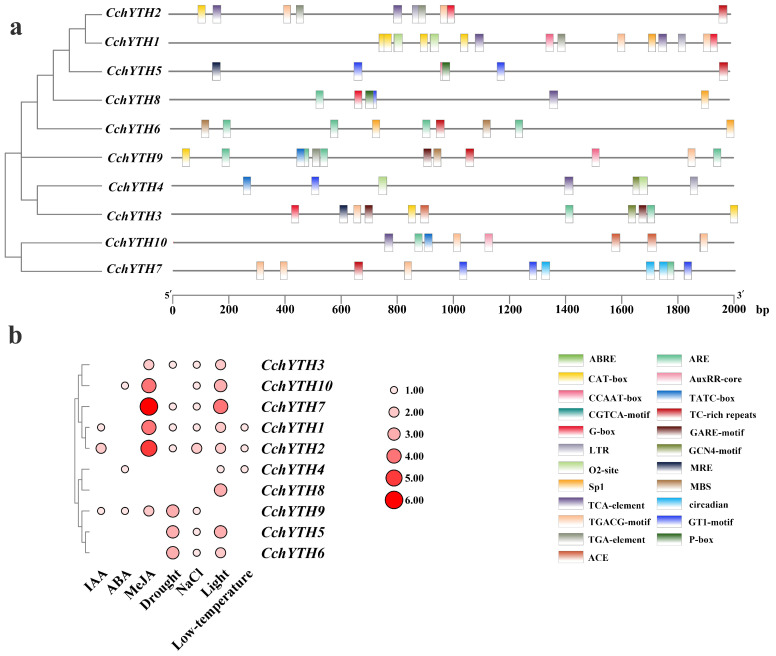
Cis-acting elements analysis of the 10 *CchYTH* gene promoters in *C. chekiangoleosa*. (**a**) Putative cis-acting element existed in the 2000 bp upstream region of *CchYTH* gene promoter. Each of the 10 predicted cis-elements is represented by a different colored box. (**b**) Number of major cis-acting elements of 10 *CchYTH* genes. Different colors and different sizes represent numbers of each cis-acting element.

**Figure 9 ijms-25-03996-f009:**
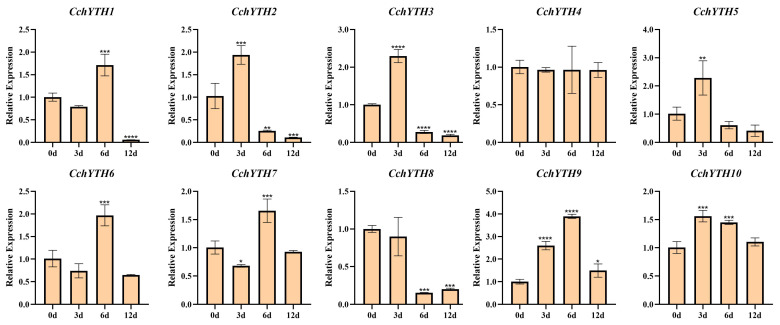
Relative expression of 10 *CchYTH*s under drought stress. The relative expression level was measured with the expression level of 0 day of treatment as the control. Different numbers of “*” indicate significant differences (* *p* < 0.05, ** *p* < 0.01, *** *p* < 0.001, **** *p* < 0.0001). Data are shown as mean ± SE, with three biological replicates.

**Table 1 ijms-25-03996-t001:** Physicochemical properties and subcellular localization of *YTH* family members in *C. chekiangoleosa*.

Gene Name	Gene ID	Number of Amino Acids	Molecular Weight (kDa)	Theoretical pI	The Instability Index	Prediction of Subcellular Localization
*CchYTH1*	Cch01T002587.1	454	50.951	5.35	38.65	cytoplasm
*CchYTH2*	Cch02T002736.1	1161	130.998	5.83	39.82	cytoplasm
*CchYTH3*	Cch02T003694.1	654	71.922	5.61	42.17	nucleus
*CchYTH4*	Cch03T002158.1	602	65.914	5.40	52.69	nucleus
*CchYTH5*	Cch05T000069.1	515	56.843	5.54	39.66	nucleus
*CchYTH6*	Cch07T000910.1	559	61.303	7.62	44.71	nucleus
*CchYTH7*	Cch08T001554.1	693	75.739	5.95	55.16	nucleus
*CchYTH8*	Cch09T004551.1	1356	150.028	5.41	43.14	cytoplasm
*CchYTH9*	Cch12T001174.1	438	48.770	9.22	29.17	nucleus
*CchYTH10*	Cch13T001809.1	386	43.564	6.17	50.25	nucleus

## Data Availability

Data is contained within the article and Appendix A.

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
