# Peer review of "Genome-Wide Identification and Expression Analysis of YTH Gene Family for Abiotic Stress Regulation in Camellia chekiangoleosa"

_ijms, 2024, doi:10.3390/ijms25073996_

Round 1

Reviewer 1 Report

Comments and Suggestions for Authors

In this study the authors identified and characterized 10 members of the YTH family genes in Camellia checkiangoleosa, which contain YTH conserved domains. They provided data about the chromosomal localization of genes and the characterization of predicted proteins.  In addition, the expression profile of genes suggested a role in the control of drought. In general, the paper is well written and can be accepted after the minor adjustment.

1.      Abbreviations should be explained when they first appear in the text: m6A

2.      Line 106: the sentence should be rewritten: “ are localiza¡ed to the nucleus and 3 (30%) localized in the cytoplasm

3.      Line 156: In my opinion it should be: Exons are parts of genes….

4.      I found misinterpretation or inconsistences in the results of section 2.6. According to the figure 6, results are not clearly described. For instance: The expression level of CchYTH2 is high in flowers, but low in other tissues. Based on the heat map, and the legend of the figure, the expression level is medium in flowers, therefore the sentence could be rephrased like:   The expression level of CchYTH2 is higher in flowers than in other tissues. The same for the expression levels of CchYTH7, CchYTH8 and CchYTH9.

5.      The last sentence of the legend of Figure 6 should be: red box indicated high expression; yellow box indicates medium expression; blue box indicates low expression.

6.      Legend of Figure 8: Exsited????

Reviewer 2 Report

Comments and Suggestions for Authors

Excellent research. Please review and address the comments in the text.

Comments on the Quality of English Language

The research is excellent. It must be accepted for publication. The authors must modify the conclusions and pay attention to other observations that appear in the text.
